# Growth Performance, Antioxidant and Immunity Capacity Were Significantly Affected by Feeding Fermented Soybean Meal in Juvenile Coho Salmon (*Oncorhynchus kisutch*)

**DOI:** 10.3390/ani13050945

**Published:** 2023-03-06

**Authors:** Qin Zhang, Fanghui Li, Mengjie Guo, Meilan Qin, Jiajing Wang, Hairui Yu, Jian Xu, Yongqiang Liu, Tong Tong

**Affiliations:** 1Guangxi Key Laboratory for Polysaccharide Materials and Modifications, Guangxi Marine Microbial Resources Industrialization Engineering Technology Research Center, School of Marine Sciences and Biotechnology, Guangxi Minzu University, 158 University Road, Nanning 530008, China; 2Key Laboratory of Biochemistry and Molecular Biology in Universities of Shandong (Weifang University), Weifang Key Laboratory of Coho Salmon Culturing Facility Engineering, Institute of Modern Facility Fisheries, Weifang University, Weifang 261061, China

**Keywords:** fermented soybean meal, coho salmon, growth performance, antioxidant, immunity, gene expression

## Abstract

**Simple Summary:**

Fish meal has been the main aquatic feed protein source for aquaculture. However, global fish meal is lacking, and the price of fish meal continues to rise, which has been unable to meet the needs. Soybean meal is currently recognized as the best choice to replace fish meal in aquatic feed, but soybean meal contains anti-nutritional factors which can affect the health of aquatic animals. Microbial fermentation is a commonly used biological method for treating soybean meal antigens and palatability. In this study, juvenile coho salmon were fed a diet with replaced 10% fish meal protein with fermented soybean meal protein supplementation for 12 weeks. The results indicated that the diet with replaced 10% fish meal protein with fermented soybean meal protein supplementation could significantly (*p* < 0.05) influence the expression of superoxide dismutase, catalase, glutathione peroxidase, glutathione S-transferase, nuclear factor erythroid 2-related factor 2, tumor necrosis factor α and interleukin-6 genes, the growth performance, the serum biochemical indices, and the activity of antioxidant and immunity enzymes.

**Abstract:**

This study aims to investigate the effects of partial dietary replacement of fish meal with unfermented and/or fermented soybean meal (fermented by *Bacillus cereus*) supplemented on the growth performance, whole-body composition, antioxidant and immunity capacity, and their related gene expression of juvenile coho salmon (*Oncorhynchus kisutch*). Four groups of juveniles (initial weight 159.63 ± 9.54 g) at 6 months of age in triplicate were fed for 12 weeks on four different iso-nitrogen (about 41% dietary protein) and iso-lipid (about 15% dietary lipid) experimental diets. The main results were: Compared with the control diet, the diet with replaced 10% fish meal protein with fermented soybean meal protein supplementation can significantly (*p* < 0.05) influence the expression of superoxide dismutase, catalase, glutathione peroxidase, glutathione S-transferase, nuclear factor erythroid 2-related factor 2, tumor necrosis factor α and interleukin-6 genes, the growth performance, the serum biochemical indices, and the activity of antioxidant and immunity enzymes. However, there was no significant effect (*p* > 0.05) on the survival rate (SR) and whole-body composition in the juveniles among the experimental groups. In conclusion, the diet with replaced 10% fish meal protein with fermented soybean meal protein supplementation could significantly increase the growth performance, antioxidant and immunity capacity, and their related gene expression of juveniles.

## 1. Introduction

Coho salmon (*Oncorhynchus kisutch*) has become one of the most promising fish in China because of its fast growth rate, high economic value, rich nutrition, containing a variety of minerals, and delicious meat [1,2,3]. At present, the feed needed by the salmon aquaculture industry is mainly fish meal, and fish meal has been the main aquatic feed protein source for aquaculture because of its high protein content, balanced amino acid composition and rich nutrition [4]. However, due to the continuous growth of the modern aquaculture industry, global fish meal is lacking, and the price of fish meal continues to rise, which has been unable to meet the needs [5]. Therefore, it is urgent to find a suitable protein source to replace fish meal in the aquaculture industry.

Soybean meal is a plant protein with high digestive protein content, wide source, and low price, so it is currently recognized as the best choice to replace fish meal in aquatic feed [6]. However, the soybean meal contains unbalanced amino acids and soybean antigen protein, urease, trypsin inhibitor, soybean lectin, phytic acid, saponins, phytoestrogens, anti-vitamins and allergens, and other anti-nutritional factors [7,8,9], which can affect the palatability, and inhibit the digestion and absorption of nutrients, and cause the damage of tissue and organ, and seriously affect the health of aquatic animals [10,11]. Microbial fermentation is a commonly used biological method for treating soybean meal antigens and palatability, and soybean meal after microbial fermentation can reduce most of the anti-nutritional factors, produce carbohydrates, digestive enzymes and other nutrients, degradation of macromolecular protein, produce small active peptides, organic acids, thereby enhancing its nutritional value and enhance the digestion and absorption of nutrients [12,13,14]. In addition, fermented soybean meal can also provide animals with probiotics, prebiotics and flavonoids and other active substances [15,16] and increase the antioxidant properties of free amino acid content and the concentration of phenolic compounds [17].

At present, there are relatively few studies on the replacement of fish meal with fermented soybean meal in coho salmon. The antibacterial substances produced by *Bacillus cereus* have the effects of promoting growth, regulating immune function, and treating diseases in livestock and poultry [18]. Therefore, coho salmon was selected as the research object, and *Bacillus cereus* was used as a fermentation strain to explore the effects of replacing part of fish meal with fermented soybean meal on the growth performance, muscle composition, antioxidant and immunity capacity, and their related gene expression of juvenile coho salmon in this study. The results provide a theoretical basis for the development and optimization of coho salmon compound feed and the healthy development of the artificial breeding industry.

## 2. Materials and Methods

### 2.1. Experimental Diets

Four different iso-nitrogen (about 41% dietary protein) and iso-lipid (about 15% dietary lipid) experimental diets were designed and based on the references [19,20,21], in which the soybean meal could replace 10% fish meal protein. The G0 diet contained 28% fish meal protein (control group). Three other diets (G1, G2 and G3) were replaced 10% fish meal protein with unfermented and/or fermented soybean meal: The G1 diet replaced by 10% unfermented soybean meal protein, the G2 diet replaced by 5% unfermented soybean meal protein and 5% fermented soybean meal protein, and the G3 diet replaced by 10% fermented soybean meal protein, based on per kg of dried feed, as shown in Table 1.

All the feed materials were provided by Conkerun Ocean Technology Co., Ltd. in Shandong, China, and they were animal food-grade. The soybean meal was fermented by *Bacillus cereus*, and the bacterial strain was collected from mangrove root soil in Maowei Sea, Qinzhou, Guangxi, China (21°81′66″ N, 108°58′46″ E). The experimental strains and fermentation conditions were derived from preliminary experiments in our lab. The inoculation amount of *Bacillus cereus* was 10% (v/m), the ratio of material to water was 1:1.4, and the fermentation was cultured at 37 °C for 60 h. The fermented soybean meal was dried for 24 h in a blast drying baker at 37 °C. A hammer mill was used to grind raw all the dry materials into a fine powder (80-μm mesh), then all the dry materials were mixed in a roller mixer for 15 min and added some water to make a hard dough. Floating pellets with a diameter of 2.0 × 3.0 mm were obtained by a single screw extruder, and they were dried in the air flow at 37 °C until the water content was below 100 g/kg. Then the dry floating pellets were sealed in plastic bags and stored at −20 °C until use.

### 2.2. Experimental Fish and Culture

Six hundred juvenile coho salmon at the age of 6 months were from a hatchery located in Benxi rainbow trout breeding farm in Liaoning, China. Outdoor feeding and breeding experiments of juvenile coho salmon were carried out at a rainbow trout breeding farm in Nanfen District, Benxi City, Liaoning, China.

After being disinfected using a concentration of 1/100,000–1/50,000 potassium permanganate, the juveniles were acclimatized for 14 days, using water temperature at 10–18 °C, water intake ≥ 100 L/s, surface velocity ≥ 2 cm/s, dissolved O_2_ ≥ 6.0 mg/L, pH 7.8–8.3 and natural light. The juveniles were fed three times a day at 08:00, 12:00 and 16:00 h, using a control diet (28% fish meal protein), and the daily feeding quantity was fed until the fish was no feeding behavior at the feeding time.

After being acclimatized for 14 days, 390 juvenile coho salmon (initial weight 159.63 ± 9.54 g) were selected for the formal experiment, and 30 of the selected juveniles were freely taken for initial samples. The remaining 360 of them were assigned randomly into 4 groups in triplicate, making a total of 12 net cages (1.0 × 1.0 × 0.8 m, L × W × H) with 30 fish in each net cage. The juveniles were cultured in the same breeding environment, and they were fed for 12 weeks using one of the 4 diets above (Table 1) and the daily feeding quantity was fed until the fish was no feeding behavior at the feeding time.

### 2.3. Sampling

The juvenile coho salmon were sampled at day 0 and the end of 12 weeks, respectively, after being starved for 24 h. All sample fish were separately anesthetized using 40 mg/L of 3-aminobenzoic acid ethyl ester methane sultanate (MS-222, Adamas Reagent, China). Then, their body weight and length were individually measured. At day 0, 20 juveniles were taken for dissecting liver samples and the other 10 juveniles for the sampling of whole fish. At the end of 12 weeks, 9 fish per net cage were randomly taken for the samples, 3 of which were for whole fish samples and 6 for the samples of serum, viscera mass, and liver.

A sterile syringe was used to collect blood from the tail vein of juvenile coho salmon; then, the blood was transferred to a 2 mL sterile enzyme-free centrifuge tube. At 3000× *g* and 4 °C, the blood was centrifuged in a centrifuge for 15 min, and the supernatant was serum. The liver weight and visceral mass weight were weighed and recorded separately for analysis of the growth performance. All the experimental samples were stored at −80 °C for subsequent analysis.

### 2.4. Calculations and Analytical Methods

#### 2.4.1. Growth Performance

The survival rate, weight gain rate, specific growth rate, condition factor, hepatosomatic index, viscerosomatic index, feed conversion ratio, and protein efficiency ratio are calculated according to the following formulas.
$$\mathrm{Survival}\;\mathrm{rate}\;(\mathrm{SR},\;\%)=100\;\times \frac{\mathrm{final}\;\mathrm{amount}\;\mathrm{of}\;\mathrm{fish}}{\mathrm{inital}\;\mathrm{amount}\;\mathrm{of}\;\mathrm{fish}}$$
$$\mathrm{Weight}\;\mathrm{gain}\;\mathrm{rate}\;(\mathrm{WGR},\;\%)=100\;\times \frac{\mathrm{final}\;\mathrm{body}\;\mathrm{weight}\;(g)\;-\;\mathrm{initial}\;\mathrm{body}\;\mathrm{weight}\;(g)}{\mathrm{initial}\;\mathrm{body}\;\mathrm{weight}\;(g)}$$
$$\mathrm{Specific}\;\mathrm{growth}\;\mathrm{rate}\;(\mathrm{SGR},\;\%/d)=100\;\times \frac{\ln(\mathrm{final}\;\mathrm{body}\;\mathrm{weight}\;(g))\;-\;\ln(\mathrm{initial}\;\mathrm{body}\;\mathrm{weight}\;(g))}{\mathrm{days}}$$
$$\mathrm{Condition}\;\mathrm{factor}\;(\mathrm{CF},\;\%)=100\;\times \frac{\;\mathrm{body}\;\mathrm{weight}\;(g)}{{\left(\mathrm{body}\;\mathrm{length}\;(\mathrm{cm})\right)}^{3}\;}$$
$$\mathrm{Hepatosomatic}\;\mathrm{index}\;(\mathrm{HSI},\;\%)=100\;\times \frac{\mathrm{liver}\;\mathrm{weight}\;(g)}{\;\mathrm{body}\;\mathrm{weight}\;(g)}$$
$$\mathrm{Viscerosomatic}\;\mathrm{index}\;(\mathrm{VSI},\;\%)=100\;\times \frac{\mathrm{viscera}\;\mathrm{weight}\;(g)}{\;\mathrm{body}\;\mathrm{weight}\;(g)}$$
$$\mathrm{Feed}\;\mathrm{conversion}\;\mathrm{ratio}\;(\mathrm{FCR})=\frac{\mathrm{total}\;\mathrm{diets}\;\mathrm{weight}\;(g)}{\;\mathrm{final}\;\mathrm{body}\;\mathrm{weight}\;(g)\;-\;\mathrm{initial}\;\mathrm{body}\;\mathrm{weight}\;(g)}$$
$$\mathrm{Protein}\;\mathrm{efficiency}\;\mathrm{ratio}\;(\mathrm{PER},\;\%)=100\;\times \frac{\mathrm{final}\;\mathrm{body}\;\mathrm{weight}\;(g)\;-\;\mathrm{initial}\;\mathrm{body}\;\mathrm{weight}\;(g)}{\;\mathrm{total}\;\mathrm{intake}\;\mathrm{of}\;\mathrm{crude}\;\mathrm{protein}\;\mathrm{weight}\;(g)}$$

#### 2.4.2. Determination of Feed and Whole Fish Composition

The compositions of feed and whole fish were analyzed following the standard methods of the Association of Official Analytic Chemists (AOAC, 2005) [22]. The samples were dried at 105 °C until constant weight in an oven to determine moisture content. The muffle furnace at 550 °C for 24 h was used to determine ash. Kjeldahl method was used to determine crude protein. Soxhlet method by ether extraction was used to determine crude lipid.

#### 2.4.3. Determination of Serum Biochemical Parameters

The indicators in serum were measured using the kit produced by Nanjing Jiancheng Bioengineering Institute (Nanjing, China) and referred to the instructions in the kit for specific operation steps. All the instructions can be found and downloaded at http://www.njjcbio.com (accessed on 1 March 2023). The total protein (TP) content was determined by the Coomassie brilliant blue method. The glucose (GLU) content was determined by the glucose oxidase method. The total cholesterol (T-CHO) content was determined by the cholesterol oxidase (COD-PAP) method. The albumin (ALB) content and alkaline phosphatase (AKP) vitality were determined by the microplate method.

#### 2.4.4. Determination of Liver Antioxidant Capacity

The indicators in the liver were measured using the kit produced by Nanjing Jiancheng Bioengineering Institute (Nanjing, China) and referred to the instructions in the kit for specific operation steps. All the instructions can be found and downloaded at http://www.njjcbio.com (accessed on 1 March 2023). The superoxide dismutase (SOD) was determined by the water-soluble tetrazole salt (WST-1) method. The catalase (CAT) was determined by the visible light method. The malondialdehyde (MDA) was determined by the thiobarbituric acid (TBA method). The total antioxidant capacity (T-AOC) was determined by the ferric-reducing ability of plasma (FRAP) method. The glutathione peroxidase (GSH-PX), glutathione S-transferase (GST), hydroxyl radical clearance ratio (OH·-CR) and superoxide radical clearance ratio (O_2_·-CR) were determined by the colorimetric method. The reduced glutathione (GSH) was determined by the microplate method.

#### 2.4.5. Expression of Antioxidant and Immunity Genes

The method of Ding et al. [23] was applied to determine the expression of *sod*, *cat*, *gsh-px*, *gst*, *nrf2*, *tnf-α* and *il-6* mRNA in the liver of the juvenile coho salmon. Briefly, the Steady Pure Universal RNA Extraction Kit and the Evo M-MLV reverse transcription kit (Accurate Biology Biotechnology Engineering Ltd., Changsha, China) were used to extract 500 ng of total RNA from samples and reverse-transcribe it into cDNA. The polymerase chain reaction (PCR) conditions were 50 °C for 30 min, 95 °C for 5 min, and 5 °C for 5 min.

The forward and reverse primers of *sod*, *cat*, *gsh-px*, *gst*, *nrf2*, *tnf-α* and *il-6* genes for reverse transcription were designed by referencing the corresponding genomic sequences of coho salmon in the National Center for Biotechnology Information (NCBI) database. The primers were synthesized by Sangon Biotech (Shanghai) Co., Ltd. (Shanghai, China). The primers were shown in Table 2, and *β-actin* was chosen as the nonregulated reference gene.

The real-time quantitative polymerase chain reaction (RT-qPCR) was conducted using an RT-qPCR System (LightCycler^®^ 96, Roche, Switzerland) and SYBR Green Pro Taq HS qPCR kit (Accurate Biology Biotechnology Engineering Ltd., Changsha, China). The RT-qPCR conditions were as follows: initial denaturation at 95 °C for 30 s, 40 cycles of denaturation at 95 °C for 5 s, annealing at 60 °C for 30 s and extension at 72 °C for 20 s.

The 2^−ΔΔCT^ method [24] was applied to calculate the relative expression levels of *sod*, *cat*, *gsh-px*, *gst*, *nrf2*, *tnf-α* and *il-6* mRNA.

### 2.5. Statistical Analysis

All the data were analyzed using IBM SPSS Statistics 25 (Chicago, IL, USA) and one-way analysis of variance (ANOVA) and tested for normality and homogeneity of variance. Duncan’s test was used for multiple comparison analysis when it was significantly different (*p* < 0.05). Statistics are expressed as means ± standard deviation (SD).

## 3. Results

### 3.1. Effect of Replacing a Portion of Fish Meal with Unfermented and/or Fermented Soybean Meal on the Growth Performance of Juvenile Coho Salmon

The WGR, SGR, CF, and PER of the juveniles in G3 and the HSI, VSI, and FCR of the juveniles in G1 and G2 were significantly higher (*p* < 0.05) than those of the juveniles in G0. The HSI, VSI, and FCR of the juveniles in G3 and the WGR, SGR, CF, and PER of the juveniles in G1 and G2 were significantly lower (*p* < 0.05) than those of the juveniles in G0. However, there was no significant difference in the SR of the juveniles between the groups (*p* > 0.05), as shown in Table 3.

### 3.2. Effect of Replacing a Portion of Fish Meal with Unfermented and/or Fermented Soybean Meal on the Whole-Body Composition of Juvenile Coho Salmon

No significant difference (*p* > 0.05) was found in the moisture, crude protein, crude lipid, and ash of juvenile coho salmon fed diets of replacement of fish meal with unfermented soybean meal and/or fermented soybean meal, as shown in Table 4.

### 3.3. Effect of Replacing a Portion of Fish Meal with Unfermented and/or Fermented Soybean Meal on the Physiological and Biochemical Indices in Serum of Juvenile Coho Salmon

The TP, GLU, ALB, AKP, and T-CHO of the juveniles in G3 were significantly higher (*p* < 0.05) than those of the juveniles in G0. The TP, GLU, ALB, AKP, and T-CHO of the juveniles in G1 and G2 were significantly lower (*p* < 0.05) than those of the juveniles in G0, as shown in Table 5.

### 3.4. Effect of Replacing a Portion of Fish Meal with Unfermented and/or Fermented Soybean Meal on the Antioxidant Capacity in the Liver of Juvenile Coho Salmon

The SOD, CAT, GSH-PX, GSH, GST, OH·-CR, O_2_·-CR, and T-AOC of the juveniles in G3, and the MDA of the juveniles in G1 and G2 were significantly higher (*p* < 0.05) than those of the juveniles in G0. The MDA of the juveniles in G3 and the SOD, CAT, GSH-PX, GSH, GST, OH·-CR, O_2_·-CR, and T-AOC of the juveniles in G1 and G2 were significantly lower (*p* < 0.05) than those of the juveniles in G0, as shown in Table 6.

### 3.5. Effect of Replacing a Portion of Fish Meal with Unfermented and/or Fermented Soybean Meal on the Expression of Antioxidant and Immune Genes in the Liver of Juvenile Coho Salmon

The expression of the *sod*, *cat*, *gsh-px*, *gst*, and *nrf2* genes in the liver of the juveniles in G3 and the expression of the *il-6* and *tnf-α* genes in the liver of the juveniles in G1 and G2 were significantly higher (*p* < 0.05) than those of the juveniles in G0. The expression of the *il-6* and *tnf-α* genes in the liver of the juveniles in G3 and the expression of *sod*, *cat*, *gsh-px*, *gst*, and *nrf2* genes in the liver of the juveniles in G1 and G2 were significantly lower (*p* < 0.05) than those of the juveniles in G0, as shown in Figure 1.

## 4. Discussion

The growth performance of fish can be used to reflect growth and health status, and it is affected by many factors, such as fish species, growth stage, nutrient deficiency, metabolic disorders, anti-nutritional factors, and toxic and harmful substances [25]. The results of this study showed that partial replacement of fish meal with fermented soybean meal could significantly increase the growth performance of juvenile coho salmon. However, partial replacement of fish meal with unfermented soybean meal could significantly decrease the growth performance of juvenile coho salmon. The reasons are supposed to be: First, unfermented soybean meal had adverse factors such as poor palatability, essential amino acid imbalance, low phosphorus utilization, high anti-nutritional factors, and easily cause lipid metabolism disorder, which will lead to decreased growth performance [26]. Second, fermented soybean meal could reduce and even eliminate anti-nutrient factors, and the protein could be degraded into easily digestible peptides or amino acids; thus, fermented soybean meal could improve the nutritional quality of feed and the digestibility of fish [27]. Third, the active bacteria, organic acids, and vitamins in fermented soybean meal would also play a positive role in growth performance [28]. Similar studies had shown that feeding largemouth bass (*Micropterus salmoides*) [21] and Macrobrachium nipponense (*Macrobrachium nipponense*) [29] with the diet with partial replacement of fish meal with fermented soybean meal significantly improved their growth performance.

Serum biochemical indexes of fish are closely related to metabolism, nutrient absorption, and health status. They are important indexes to evaluate physiology and pathology and are widely used to measure metabolism and health status [30,31]. TP and ALB in the blood are synthesized by the liver, and the increase of TP and ALB content indicates that the ability of the liver to synthesize protein is enhanced. AKP is one of the important indicators of fish physiological activity and disease diagnosis, which can reflect the anti-stress ability of biological organisms [32]. T-CHO is an important index to reflect the body’s lipid metabolism [33]. GLU is the main functional substance of the body, and its content is affected by nutrition and feed intake [34]. The results of this study showed that partial replacement of fish meal with fermented soybean meal could significantly increase the serum biochemical indexes of juvenile coho salmon, indicating that fermented soybean meal could be used as a protein substitute for fish meal to improve the health of juvenile coho salmon. The reasons are supposed to be: First, fermented soybean meal could improve the intestinal structure and function of fish, increase the activity of digestive enzymes, and increase the absorption and utilization of dietary proteins and lipids [35]. Second, compared with macromolecular proteins, the small peptides in fermented soybean meal are more easily absorbed by fish, which could improve the diet protein utilization rate, consequently enhancing the serum protein content of fish [12]. Third, fermented soybean meal could decrease the content of soybean saponins, increase the activity of α-glucosidase, and improve the absorption of glucose [36]. Fourth, fermented soybean meal could not only reduce the inhibitory effect of soy isoflavones on serum T-CHO levels but also stimulate the antioxidant system of the body, thereby inhibiting the process of lipid oxidation and increasing the content of T-CHO in the serum [37]. In addition, bioactive peptides during fermentation can act as immune stimulants to enhance AKP activity [38].

Nuclear factor erythroid 2-related factors (*nrf2*) is an important nuclear transcription factor and can be involved in a variety of cellular processes, including maintaining intracellular redox balance, cell proliferation/differentiation, metabolism, protein homeostasis and inflammation regulation, and disease development [39,40]. The activation of the *nrf2* signaling pathway can initiate the expression of multiple downstream target proteins, such as SOD, CAT, GPX, glutathione ligase (γ-GCS), glutathione catalase (GR), glutathione S-transferase (GST) and glucose-6-phosphate kinase (G-6-PDH) [41]. The expression of these genes is an important way for the body to resist oxidative stress damage [42]. *Nrf2* signaling pathway can negatively regulate various cytokines (TNF-α, IL-1 and IL-6), chemokines, cell adhesion factors, matrix metalloproteinases, cyclooxygenase-2, inducible nitric oxide synthase, and other inflammatory mediators, which plays a protective role in the dysfunction caused by inflammation [43]. IL-6 and TNF-α are often used as indicators of the inflammatory response [44]. MDA content has been used by many researchers to evaluate the effect of protein replacement sources on the antioxidant capacity of fish, which can be used as an important marker of endogenous oxidative damage in organisms [45]. The results of this study showed that partial replacement of fish meal with fermented soybean meal could significantly increase the antioxidant capacity and the expression of their related gene in the liver and significantly decrease the expression of *il-6* and *tnf-α* gene in the liver of juvenile coho salmon. However, partial replacement of fish meal with unfermented soybean meal could significantly decrease the antioxidant capacity and the expression of their related gene in the liver and significantly increase the expression of the *il-6* and *tnf-α* genes in the liver of juvenile coho salmon. The reasons are supposed to be: First, the soybean globulin and β-conglycinin in soybean meal could destroy the antioxidant system of fish and cause oxidative damage [46]. Previous studies have shown that soybean meal in feed may cause oxidative stress in fish such as gilthead sea bream (*Sparus aurata*) [47]. Second, a high concentration of soybean peptides and phenols in fermented soybean meal could up-regulate *nrf2* gene expression, induce the expression of the *sod*, *cat*, *gsh*, and *gsh-px* genes, and improve the antioxidant ability of the body [48,49]. Lee et al. found that an appropriate proportion of fermented soybean meal in a diet can increase the activities of SOD, GSH-Px, and GSH in the liver [50]. Third, *Bacillus* could stimulate the production of antioxidant enzymes and antioxidants, thereby scavenging free radicals, maintaining homeostasis, improving antioxidant capacity, and activating the Nrf2 pathway [51]. Fourth, the replacement of fish meal protein with 10% fermented soybean meal protein was insufficient for causing a change in the body’s ability to recognize foreign bodies and did not lead to an inflammatory reaction [52]. In addition, after soybean meal fermentation, a unique fragrance could be formed, which can promote the feeding of aquatic animals and increase their immunity [53].

However, the results of this study showed that partial replacement of fish meal with unfermented and/or fermented soybean meal had no significant effect on the survival rate and whole-body composition of juvenile coho salmon. The reasons are supposed to be: First, the energy required by fish to maintain normal life activities mainly depends on the breakdown of protein and fat, and fish meal contains a complete set of essential amino acids that meet the protein requirements of most aquatic animals [54,55]. Second, the crude protein and crude fat contents of the four diets in this study were the same and were enough to satisfy the daily needs of juvenile coho salmon. Third, fish body composition is affected by external conditions such as feed nutrients, food composition, aquaculture water environment and season, but fish body composition was not affected by plant protein levels [56]. Similar results were obtained in pompano (*Trachinotus ovatus*) [53] and Florida pompano (*Trachinotus carolinus*) [56] fed with fermented soybean meal partially replacing fish meal. However, studies have shown that a high proportion of fermented soybean meal instead of fish meal significantly increased the whole-body moisture and reduced crude protein and crude lipid content of Japanese seabass (*Lateolabrax japonicus*) [57]. In giant grouper (*Epinephelus lanceolatus*), high levels of fermented soybean meal replacement also significantly increased whole-fish moisture and decreased crude protein and crude lipid content [58]. The above inconsistent results might be related to the strains of fermented soybean meal, the basic feed formula, the substitution ratio of fermented soybean meal, the types of aquatic animals, the breeding cycle, and the growth stage.

## 5. Conclusions

In conclusion, the diet with replaced 10% fish meal protein with fermented soybean meal protein supplementation can significantly influence the expression of superoxide dismutase, catalase, glutathione peroxidase, glutathione S-transferase, nuclear factor erythroid 2-related factor 2, tumor necrosis factor α and interleukin-6 genes, the growth performance, the serum biochemical indices, and the activity of antioxidant and immunity enzymes of juvenile coho salmon. The results provide a theoretical basis for the development and optimization of coho salmon compound feed and the healthy development of the artificial breeding industry.

## Figures and Tables

**Figure 1 animals-13-00945-f001:**
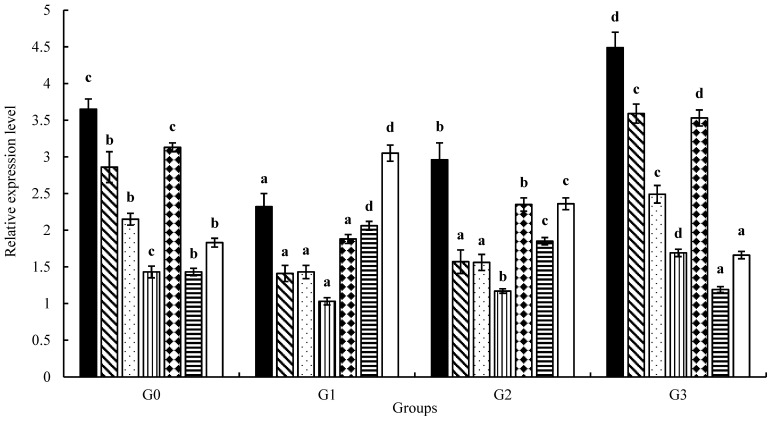
Effect of fish meal replaced by unfermented and/or fermented soybean meal on the expression levels of the antioxidant and immunity genes in the liver of juvenile coho salmon, in which (■) indicates superoxide dismutase (*sod*), (
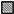
) catalase (*cat*), (
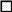
) glutathione peroxidase (*gsh-px*), (
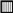
) glutathione S-transferase (*gst*), (
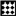
) nuclear factor erythroid 2-related factor 2 (*nrf2*), (
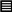
) tumor necrosis factor α (*tnf-α*), and (□) interleukin-6 (*il-6*). All above data are mean ± SD (*n* = 3 × 3 × 3), and different superscript letters indicate significant differences among the data (*p* < 0.05).

**Table 1 animals-13-00945-t001:** Experimental diet formula (g/kg of dried feed) and approximate composition (%, dry matter percentage).

Ingredients	G0	G1	G2	G3
Fish meal ^1^	401.00	258.00	258.00	258.00
Soybean meal ^2^	0.00	213.60	106.80	0.00
Fermented soybean meal ^3^	0.00	0.00	90.50	181.10
Chicken powder ^4^	100.00	100.00	100.00	100.00
Shrimp powder ^5^	100.00	100.00	100.00	100.00
Wheat middling ^6^	178.80	178.40	178.40	178.40
Starch	30.20	30.20	30.20	30.20
Cellulose	85.30	0.00	16.30	32.50
Fish oil	40.10	55.30	55.30	55.30
Soybean oil	40.10	40.00	40.00	40.00
Ca(H_2_PO_4_)_2_	10.10	10.10	10.10	10.10
Mineral premix ^7^	5.20	5.20	5.20	5.20
Vitamin premix ^8^	5.20	5.20	5.20	5.20
Choline	3.00	3.00	3.00	3.00
Vitamin C	1.00	1.00	1.00	1.00
Approximate composition				
Crude protein	41.78	41.42	41.38	41.26
Fish meal protein	28.00	18.00	18.00	18.00
Soybean meal protein	0.00	10.00	5.00	0.00
Fermented soybean meal protein	0.00	0.00	5.00	10.00
Crude lipid	15.21	15.22	15.21	15.20

Note: ^1^ Fish meal: protein content 70.00%, lipid content 8.00%. ^2^ Soybean meal: protein content 46.81%, lipid content 1.84%. ^3^ Fermented soybean meal: protein content 55.21%, lipid content 1.93%. ^4^ Chicken powder: protein content 62.00%, lipid content 12.00%. ^5^ Shrimp powder: protein content 49.00%, lipid content 8.00%. ^6^ Wheat middling: protein content 11.00%, lipid content 1.60%. ^7^ Composition (mg/kg mineral premix): AlK(SO_4_)_2_·12H_2_O, 123.7; CaCl_2_, 17,879.8; CuSO_4_·5H_2_O, 31.7; CoCl_2_·6H_2_O, 48.9; FeSO_4_·7H_2_O, 707.4; MgSO_4_·7H_2_O, 4316.8; MnSO_4_·4H_2_O, 31.1; ZnSO_4_·7H_2_O, 176.7; KCl, 1191.9; KI, 5.3; NaCl, 4934.5; Na_2_SeO_3_·H_2_O, 3.4; Ca(H_2_PO_4_)_2_·H_2_O, 12,457.0; KH_2_PO_4_, 9930.2. ^8^ Composition (IU or g/kg vitamin premix): retinal palmitate, 10,000 IU; cholecalciferol, 4000 IU; α-tocopherol, 75.0 IU; menadione, 22.0 g; thiamine HCl, 40.0 g; riboflavin, 30.0 g; D-calcium pantothenate, 150.0 g; pyridoxine HCl, 20.0 g; meso-inositol, 500.0 g; D-biotin, 1.0 g; folic acid, 15.0 g; ascorbic acid, 200.0 g; niacin, 300.0 g; cyanocobalamin, 0.3 g.

**Table 2 animals-13-00945-t002:** Real-time quantitative PCR primers for genes of coho salmon.

Gene	Primer Sequence	GenBank	Tm (°C)	Size (bp)
*β-actin* ^1^	F: CCAAAGCCAACAGGGAGAA R: AGGGACAACACTGCCTGGAT	BG933897	60	91
*Sod* ^2^	F: CCGTTGGTGTTGTCTCCGAAGG R: GAGGGTGACAATGCTCCAGTGAAG	XM_014198383	60	101
*gsh-px* ^3^	F: GATTCGTTCCAAACTTCCTGCTA R: GCTCCCAGAACAGCCTGTTG	BG934453	60	140
*gst* ^4^	F: CGCATTGACATGATGTGTGA R: TGTCGAGGTGGTTAGGAAGG	DQ367889	60	121
*cat* ^5^	F: GCGTTCGGGTACTTTGAGGTGAC R: TGGAGAAGCGGATGGCGATAGG	BG935638	60	103
*nrf2* ^6^	F: TAGAGACGAGCAGCGAGCCAAG R: GTTGAAGTCATCCACAGGCAGGTC	NM_001139807	60	82
*il-6* ^7^	F: GAGCTACGTAACTTCCTGGTTGAC R: GCAAGTTTCTACTCCAGGCCTGAT	XM_014143031	60	129
*tnf-α* ^8^	F: GGCGAGCATACCACTCCTCT R: TCGGACTCAGCATCACCGTA	AY848945	60	124

Note: ^1^
*β-actin*: Reference gene. ^2^
*sod*: Superoxide dismutase gene. ^3^
*gsh-px*: Glutathione peroxidase gene. ^4^
*gst*: Glutathione S-transferase gene. ^5^
*cat*: Catalase gene. ^6^
*nrf2*: Nuclear factor erythroid 2-related factor 2 gene. ^7^
*il-6*: Interleukin-6 gene. ^8^
*tnf-α*: Tumor necrosis factor α gene.

**Table 3 animals-13-00945-t003:** Effect of fish meal replaced by unfermented and/or fermented soybean meal on growth performance of the juvenile coho salmon.

	G0	G1	G2	G3
Initial weight (g)	159.63 ± 9.54	159.63 ± 9.54	159.63 ± 9.54	159.63 ± 9.54
Final weight (g)	583.49 ± 10.97 ^c^	473.01 ± 12.16 ^a^	545.08 ± 6.09 ^b^	617.07 ± 4.28 ^d^
SR ^1^ (%)	93.2 ± 1.73	91.2 ± 2.11	92.2 ± 1.24	94.6 ± 1.21
WGR ^2^ (%)	265.53 ± 6.87 ^c^	196.32 ± 7.68 ^a^	241.46 ± 3.81 ^b^	286.57 ± 2.68 ^d^
SGR ^3^ (%/d)	1.54 ± 0.02 ^c^	1.29 ± 0.03 ^a^	1.46 ± 0.01 ^b^	1.60 ± 0.01 ^d^
CF ^4^ (%)	1.91 ± 0.02 ^c^	1.42 ± 0.02 ^a^	1.67 ± 0.01 ^b^	2.07 ± 0.02 ^d^
HIS ^5^ (%)	1.58 ± 0.03 ^b^	1.71 ± 0.03 ^c^	1.68 ± 0.02 ^c^	1.46 ± 0.02 ^a^
VSI ^6^ (%)	11.67 ± 0.31 ^b^	12.75 ± 0.64 ^c^	12.51 ± 0.35 ^c^	10.24 ± 0.43 ^a^
FCR ^7^	1.64 ± 0.03 ^b^	1.95 ± 0.03 ^d^	1.86 ± 0.02 ^c^	1.53 ± 0.02 ^a^
PER ^8^ (%)	231.31 ± 7.42 ^c^	181.93 ± 3.22 ^a^	208.44 ± 6.85 ^b^	252.19 ± 5.31 ^d^

Note: All above data are mean ± SD (*n* = 3 × 3 × 3) except SR is mean ± SD (*n* = 30 × 3), and different superscript letters in the same row indicate significant differences among the data (*p* < 0.05). ^1^ SR: Survival rate. ^2^ WGR: Weight gain rate. ^3^ SGR: Specific growth rate. ^4^ CF: Condition factor. ^5^ HSI: Hepatosomatic index. ^6^ VSI: Viscera index. ^7^ FCR: Feed conversion ratio. ^8^ PER: Protein efficiency ratio.

**Table 4 animals-13-00945-t004:** Effect of fish meal replaced by unfermented and/or fermented soybean meal on whole body composition of the juvenile coho salmon (%/per g of wet weight).

	G0	G1	G2	G3
Moisture	75.22 ± 0.19	75.27 ± 0.10	75.38 ± 0.68	75.04 ± 0.36
Crude protein	18.25 ± 0.17	17.67 ± 0.57	17.89 ± 0.81	17.98 ± 0.44
Crude lipid	5.36 ± 0.39	5.28 ± 0.34	5.30 ± 0.09	5.34 ± 0.13
Ash	1.45 ± 0.07	1.50 ± 0.06	1.67 ± 0.10	1.52 ± 0.09

Note: All above data are mean ± SD (*n* = 3 × 3 × 3), and different superscript letters in the same row indicate significant differences among the data (*p* < 0.05).

**Table 5 animals-13-00945-t005:** Effect of fish meal replaced by unfermented and/or fermented soybean meal on serum physiological and biochemical indices of the juvenile coho salmon.

	G0	G1	G2	G3
TP ^1^ (g/L)	54.46 ± 0.17 ^c^	36.21 ± 0.21 ^a^	44.98 ± 0.06 ^b^	60.28 ± 1.34 ^d^
GLU ^2^ (mmol/L)	5.06 ± 0.35 ^c^	2.97 ± 0.14 ^a^	4.08 ± 0.33 ^b^	5.87 ± 0.12 ^d^
ALB ^3^ (g/L)	38.79 ± 2.37 ^c^	19.23 ± 0.92 ^a^	30.3 ± 2.76 ^b^	44.27 ± 2.79 ^d^
AKP ^4^ (U/mL)	20.36 ± 1.25 ^c^	8.26 ± 1.20 ^a^	13.19 ± 1.52 ^b^	23.69 ± 0.92 ^d^
T-CHO ^5^ (mmol/L)	7.77 ± 0.33 ^c^	3.19 ± 0.07 ^a^	4.94 ± 0.87 ^b^	8.54 ± 0.11 ^d^

Note: All above data are mean ± SD (*n* = 3 × 3 × 3), and different superscript letters in the same row indicate significant differences among the data (*p* < 0.05). ^1^ TP: Total protein. ^2^ GLU: Glucose. ^3^ ALB: Albumin. ^4^ AKP: Alkaline phosphatase. ^5^ T-CHO: Total cholesterol.

**Table 6 animals-13-00945-t006:** Effect of fish meal replaced by unfermented and/or fermented soybean meal on the antioxidant capacity in the liver of the juvenile coho salmon.

	G0	G1	G2	G3
SOD ^1^ (U/mg)	822.5 ± 32.71 ^b^	627.12 ± 32.84 ^a^	642.77 ± 46.08 ^a^	977.36 ± 54.18 ^c^
CAT ^2^ (U/mg)	318.75 ± 15.31 ^b^	221.77 ± 20.15 ^a^	257.31 ± 22.94 ^a^	365.02 ± 16.37 ^c^
GSH-PX ^3^ (U/mg)	19.48 ± 2.48 ^c^	9.09 ± 1.5 ^a^	13.62 ± 1.44 ^b^	23.27 ± 1.81 ^d^
GSH ^4^ (U/mg)	118.13 ± 9.42 ^c^	68.32 ± 4.35 ^a^	88.56 ± 11.77 ^b^	153.86 ± 16.24 ^d^
GST ^5^ (U/mg)	41.86 ± 3.28 ^c^	26.62 ± 2.25 ^a^	34.71 ± 2.86 ^b^	50.52 ± 4.96 ^d^
MDA ^6^ (mmol/g)	3.82 ± 0.25 ^b^	6.13 ± 0.33 ^d^	4.95 ± 0.27 ^c^	3.12 ± 0.27 ^a^
OH·-CR ^7^ (U/g)	102.12 ± 9.15 ^c^	42.86 ± 3.98 ^a^	75.54 ± 6.21 ^b^	136.96 ± 11.75 ^d^
O_2_·-CR ^8^ (U/g)	70.12 ± 3.18 ^c^	34.84 ± 1.75 ^a^	51.43 ± 2.37 ^b^	84.27 ± 4.41 ^d^
T-AOC ^9^ (mmol/g)	2.26 ± 0.07 ^c^	1.27 ± 0.15 ^a^	1.67 ± 0.18 ^b^	2.73 ± 0.12 ^d^

Note: All above data are mean ± SD (*n* = 3 × 3 × 3), and different superscript letters in the same row indicate significant differences among the data (*p* < 0.05). ^1^ SOD: superoxide dismutase. ^2^ CAT: catalase. ^3^ GSH-PX: glutathione peroxidase. ^4^ GSH: glutathione. ^5^ GST: glutathione S-transferase. ^6^ MDA: malondialdehyde. ^7^ OH·-CR: hydroxyl radical clearance ratio. ^8^ O_2_·-CR: superoxide radical clearance ratio. ^9^ T-AOC: total antioxidant capacity.

## Data Availability

The data that support the findings of this study are available from the corresponding author upon reasonable request.

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
