# Peer review of "Growth Performance, Antioxidant and Immunity Capacity Were Significantly Affected by Feeding Fermented Soybean Meal in Juvenile Coho Salmon (Oncorhynchus kisutch)"

_animals, 2023, doi:10.3390/ani13050945_

Round 1

Reviewer 1 Report

Review of ‘Growth Performance, Antioxidant and Immunity Capacity, and Their Related Gene Expression were Significantly Affected by Feeding Partial Replacement of Fish Meal with Soybean Meal Fermented by Bacillus Cereus in Juvenile Coho Salmon (Oncorhynchus kisutch)’ by Qin Zhang, Fanghui Li, Mengjie Guo, Meilan Qin, Jiajing Wang, Hairui Yu, Jian Xu, Yongqiang Liu, Tong Tong.

The authors should shorten and modify the title. It is too long and overloaded with details

The Simple Summary is mandatory for this journal, but this section is missing.

Abstract

The current abstract is wordy and overloaded with unnecessary information. The authors should reduce this section.

Introduction

Pg 1 Ln 23: Suggest changing ‘with 6 months of age’ to ‘at 6 months of age’

Pg 2 Ln 52: Suggest changing ‘continuous growth of modern’ to ‘continuous growth of the modern’

Pg 1 Ln 55: Suggest changing ‘aquaculture industry’ to ‘the aquaculture industry’

Pg 1 Ln 72: Suggest changing ‘fish meal by’ to ‘fish meal with’

Methods

Pg 3 Ln 112: Suggest changing ‘into fine powder’ to ‘into a fine powder’

Pg 3 Ln 113: Please, clarify ‘added few waters’ 

Pg 3 Ln 118: Suggest changing ‘salmon with age’ to ‘salmon at age’

Pg 3 Ln 120: Suggest changing ‘were carried out at rainbow trout breeding farm’ to ‘were carried out at a rainbow trout breeding farm’

Pg 3 Ln 123: Suggest changing ‘water temperature with’ to ‘water temperature at’

Pg 3 Ln 128: Suggest changing ‘390  of  juvenile  coho  salmon’ to ‘390  juvenile  coho  salmon’

Pg 4 Ln 133: Suggest changing ‘using one of 4 diets above’ to ‘using one of the 4 diets above’

Pg 4 Ln 136: Suggest changing ‘at 0 day’ to ‘at day 0’ and throughout the text

Pg 4 Ln 139: Suggest changing ‘At 0 day, 20 of’ to ‘At day 0, 20’

Pg 4 Ln 140: Suggest changing ‘another 10 of juveniles’ to ‘ the other 10 juveniles’

Pg 4 Ln 142: Suggest changing ‘in which 3’ to ‘3 of which’

Pg 5 Ln 164: Suggest changing ‘were used to determine ash’ to ‘was used to determine ash’

Pg 5 Ln 170: Suggest changing ‘Coomassie brilliant blue method. The glucose (GLU) content was determined by glucose oxidase method. The total cholesterol (T-CHO) content was determined by cholesterol oxidase (COD-PAP) method. The albumin (ALB) content and alkaline phosphatase (AKP) vitality were determined by microplate method’ to ‘the Coomassie brilliant blue method. The glucose (GLU) content was determined by the glucose oxidase method. The total cholesterol (T-CHO) content was determined by the cholesterol oxidase (COD-PAP) method. The albumin (ALB) content and alkaline phosphatase (AKP) vitality were determined by the microplate method’

Pg 5 Ln 168-174. Please, provide references for these methods.

Pg 5 Ln 176: Suggest changing ‘in liver’ to ‘in the liver’ here and throughout the text

Pg 6 Ln 209: Suggest changing ‘was was applied’ to ‘was applied’

Results

Pg 6 Ln 223: Suggest changing ‘difference on’ to ‘difference in’

Pg 6 Ln 217, 232: Suggest changing ‘portion fish meal’ to ‘portion of fish meal’

Pg 7 Ln 241: Suggest changing ‘portion fish meal’ to ‘portion of fish meal’

Pg 7 Ln 253: Suggest changing ‘in liver’ to ‘in the liver’

Pg 8 Ln 266: Suggest changing ‘portion fish meal’ to ‘portion of fish meal’

Pg 8 Ln 277: Suggest changing ‘significant difference’ to ‘significant differences’

Pg 9 Ln 285, 289: Suggest changing ‘significant difference’ to ‘significant differences’

Pg 10 Ln 297, 302: Suggest changing ‘significant difference’ to ‘significant differences’

Pg 11 Ln 309, 313: Suggest changing ‘significant difference’ to ‘significant differences’

Figures 1-7: Suggest combining these figures into one.

Discussion

Pg 12 Ln 319: Suggest changing ‘but which is affected’ to ‘and it is affected’

Pg 12 Ln 348: Suggest changing ‘the health in’ to ‘the health of’

Pg 13 Ln 385: Suggest changing ‘Previous studies had shown’ to ‘Previous studies have shown’

Pg 13 Ln 394: Suggest changing ‘meal protein by’ to ‘meal protein with’

Pg 13 Ln 396: Suggest changing ‘and not lead’ to ‘and did not lead’

Pg 13 Ln 400: Suggest changing ‘were no significant effect on survival rate’ to ‘had no significant effect on the survival rate’

Pg 13 Ln 411: Suggest changing ‘had shown that high proportion’ to ‘have shown that a high proportion’

Conclusion

This section is too short and should be updated with the main findings and ways for further research in this field.

Reviewer 2 Report

This manuscript does a good job of pointing out the effects of unfermented and/or fermented soybean meal on juvenile coho salmon. The experimental design is sound, and the results are also reliable. Still some improvements have to be done in the manuscript to enhance your conclusion and strengthen the argument. After mirror revision this manuscript can be considered for publication in Animals.

  1. Table 1 The total feed weight of G1, G2 and G3 are 999.9 g, less than 1000g. Please check it again.

2. Line 106 “Maaowei sea” Please check the spelling of the place according to the longitude and latitude.

3.Line125"natural pH and light"how to understand the natural pH? 

4.Line 139-140"At 0 day, 20 of juveniles were taken for dissecting samples of serum, viscera mass, and liver...." what did you do use these sample? The data didn't present in the manuscript.

5.Line 218 Coho Salmon: The first letter does not need to be capitalized. Please check the rest of the text. Line 309, (P < 0.05), P is in italics. A lot of  little problems, please check them.

6.Table 4 There is no significant difference between the groups, so the superscript letters in the same row should be deleted.

7."data are mean ± SD (n = 3 × 3 × 3)"Please explain why the n means  "3 × 3 × 3".

8. Line 391-392 The authors talked about the function of "Bacillus". The author use the Bacillus to ferment the soybean meal.When you used the fermented soybean meal as the gredients and made the diet, the Bacillus is live or die? 
